# Anticancer Effect of Heparin–Taurocholate Conjugate on Orthotopically Induced Exocrine and Endocrine Pancreatic Cancer

**DOI:** 10.3390/cancers13225775

**Published:** 2021-11-18

**Authors:** Hae Hyun Hwang, Hee Jeong Jeong, Sangwu Yun, Youngro Byun, Teruo Okano, Sung Wan Kim, Dong Yun Lee

**Affiliations:** 1Department of Bioengineering, College of Engineering, Hanyang University, Seoul 04763, Korea; haehyun87@gmail.com (H.H.H.); eljvs@naver.com (H.J.J.); yunsangwu@naver.com (S.Y.); 2Research Institute of Pharmaceutical Sciences, College of Pharmacy, Seoul National University, Seoul 08826, Korea; yrbyun@snu.ac.kr; 3Cell Sheet Tissue Engineering Center (CSTEC), Department of Pharmaceutics and Pharmaceutical Chemistry, College of Pharmacy, University of Utah, Salt Lake City, UT 84112, USA; teruo.okano@utah.edu; 4Center for Controlled Chemical Delivery, Department of Pharmaceutics and Pharmaceutical Chemistry, University of Utah, Salt Lake City, UT 84112, USA; SW.Kim@pharm.utah.edu; 5Institute of Nano Science & Technology (INST), Hanyang University, Seoul 04763, Korea; 6Elixir Pharmatech Inc., Seoul 04763, Korea

**Keywords:** heparin taurocholate conjugate, pancreatic cancer cell, anticancer, antiangiogenesis, orthotopic animal

## Abstract

**Simple Summary:**

Pancreatic cancer has a less than 9% 5-year survival rate among patients because it is very difficult to detect and diagnose early. Combinatorial chemotherapy with surgery or radiotherapy is a potential remedy to treat pancreatic cancer. However, these strategies still have side effects such as hair loss, skin soreness and fatigue. To overcome these side effects, angiogenesis inhibitors such as sunitinib are used to deliver targeted blood vessels around tumor tissues, including pancreatic cancer tumors. It is still controversial whether antiangiogenesis therapy is sufficient to treat pancreatic cancer. So far, many scientists have not been focused on the tumor types of pancreatic cancer when they have developed antipancreatic cancer medication. Here, we used heparin–taurocholate (LHT) as an anticancer drug to treat pancreatic cancer through inhibition of angiogenic growth factors. In this study, we examined the anticancer efficacy of LHT on various types of pancreatic cancer in an orthotopic model.

**Abstract:**

Pancreatic cancers are classified based on where they occur, and are grouped into those derived from exocrine and those derived from neuroendocrine tumors, thereby experiencing different anticancer effects under medication. Therefore, it is necessary to develop anticancer drugs that can inhibit both types. To this end, we developed a heparin–taurocholate conjugate, i.e., LHT, to suppress tumor growth via its antiangiogenic activity. Here, we conducted a study to determine the anticancer efficacy of LHT on pancreatic ductal adenocarcinoma (PDAC) and pancreatic neuroendocrine tumor (PNET), in an orthotopic animal model. LHT reduced not only proliferation of cancer cells, but also attenuated the production of VEGF through ERK dephosphorylation. LHT effectively reduced the migration, invasion and tube formation of endothelial cells via dephosphorylation of VEGFR, ERK1/2, and FAK protein. Especially, these effects of LHT were much stronger on PNET (RINm cells) than PDAC (PANC1 and MIA PaCa-2 cells). Eventually, LHT reduced ~50% of the tumor weights and tumor volumes of all three cancer cells in the orthotopic model, via antiproliferation of cancer cells and antiangiogenesis of endothelial cells. Interestingly, LHT had a more dominant effect in the PNET-induced tumor model than in PDAC in vivo. Collectively, these findings demonstrated that LHT could be a potential antipancreatic cancer medication, regardless of pancreatic cancer types.

## 1. Introduction

Pancreatic cancer is an obstinate disease that is the fourth-leading cause of cancer death in the United States. Surgery is possible for only 20% of patients because the pancreas is deep in the abdomen and surrounded by other organs, and because detecting pancreatic cancer for early diagnosis is difficult. In addition, treatment of every pancreatic cancer patient is different due to the nature of pancreatic cancer.

There are two types of pancreatic cancer, pancreatic ductal adenocarcinoma (PDAC) and pancreatic neuroendocrine tumor (PNET) (Appendix A). PDAC is a type of exocrine pancreatic cancer that differentiates from small tubes called ducts consisting of duct cells and acinar cells in the pancreas. PDAC is normally an aggressive and malignant cancer of the pancreas, with a 5-year survival rate of only 9% [1], and is a major subtype that accounts for more than 90% of all pancreatic cancers [2]. This high aggressiveness is because PDAC is quite invasive, leading to distant metastases at an early stage of the disease. In addition, the progression of the cancer is already significant at the time of diagnosis, and the transition to another organ such as the liver or lymph node can occur [3]. On the other hand, PNETs are a group of tumors that differentiate from neuroendocrine cells or islet of Langerhans cells. PNETs are much less common and have a better prognosis, as they are slow-growing tumors compared with ductal adenocarcinomas. However, neuroendocrine tumor cells are more frequently differentiated than PDACs, and fast-growing pancreatic neuroendocrine tumors also exist. In addition, PNET cells affect the endocrine part of the pancreas, specifically the islet cells, which causes problems with the secretion of hormones such as insulin, glucagon and amylin in the body [4].

Combinatorial chemotherapy with surgery or radiotherapy is a potential remedy to treat pancreatic cancer. Recently, glycogen synthase kinase 3 β (GSK3β) has emerged as a new potential target in PDAC due to its involvement in tumor-promoting properties and chemoresistance [5,6,7]. To target GSK3β, its inhibitors such as ATP competitive and non-ATP competitive are developed [8,9,10,11,12]. In addition, cyclin-dependent kinase 1 (CDK1), a stimulator of the cell cycle, can be targeted to treat patients with PDAC [13]. To target PDAC diagnosis and therapy, a computational method of predicting the related gene expression and antidrug is also developed [14]. Chemotherapy is used to treat progressive tumor cells located in pancreas because they remain without being completely removed by surgery. However, these strategies still have side effects such as hair loss, skin soreness and fatigue due to the damage of normal cells as well as cancer cells. To overcome these side effects, angiogenesis inhibitors such as Sunitinib (marketed as Sutent^®^ by Pfizer) are used to be transferred around tumor tissues. This process effectively attenuates proliferating endothelial cells without affecting normal cells [15,16,17,18,19,20,21]. In most solid cancers, the vascular endothelial growth factor (VEGF) family plays an important role in its angiogenic properties. VEGF binds to a VEGF receptor, which belongs to the class of tyrosine kinase receptors, and promotes angiogenesis, which causes tumor progression and poor prognosis [22]. Sutent^®^ inhibits VEGF-mediated angiogenesis by acting as a receptor tyrosine kinases (RTKs) inhibitor at the tumor site. However, it is still controversial whether antiangiogenesis therapy is sufficient to treat pancreatic cancer or not [16,17,18].

Heparin is commonly used as an anticoagulant, but it can be also used as an anticancer drug based on its antiangiogenic properties [23]. The antiangiogenic activity of heparin is attributed to its binding to VEGF and the subsequent inhibition of VEGF receptor (VEGFR) phosphorylation [24]. In addition, heparin can significantly inhibit tumor cell adhesion through P-selectin mediation by the sulfate groups on its glucosamine residue [25,26]. However, its dose should be critically monitored because its anticoagulant properties cause side effects such as hemorrhage and heparin-induced thrombocytopenia (HIT). Therefore, to reduce the side effects of heparin for clinical applications, chemically modified heparins have been developed [27,28,29,30,31,32,33]. Previously, we developed a low-molecular-weight heparin–taurocholate conjugate (LHT) that consists of seven taurocholate molecules conjugated to the carboxylic groups of low-molecular-weight heparin (LMWH) [32,34,35] (Appendix A). The conjugated taurocholate, one of the bile acids, could not only reduce the anticoagulant activity of heparin but also improve the stability of heparin itself via the formation of a polyproline-type helical structure [27,31]. In fact, the anticoagulant activity of LHT was 12% that of unmodified heparin itself [31,34]. Therefore, when LHT was intravenously infused in beagle dogs, there was no safety issue [35,36]. In addition, this LHT could be orally absorbable through the binding of taurocholate in the LHT with the bile acid receptors in the intestinal cells of preclinical animal models, with higher safety [35,36]. However, since clinical trials using this LHT have not yet been conducted, further investigation related to hemorrhage at the clinical level is needed [37].

Until now, there are still no reports that heparin or heparin derivatives can suppress different types of pancreatic cancer in an orthotopic model. It is possible that orally absorbable LHT can inhibit or attenuate the growth of pancreatic cancers without any adverse effect. Therefore, in this study we first examined the inhibitory effects of LHT against different kinds of pancreatic cancer using an orthotopic model. We compared the differences in LHT effects on three kinds of pancreatic cancer and investigated its antiangiogenic properties through in vitro and in vivo experiments.

## 2. Materials and Methods

### 2.1. Preparation of Heparin–Taurocholate Conjugate (LHT)

Low-molecular-weight heparin–taurocholate conjugate (LHT) was synthesized via amide linkage, which was previously described [31]. A primary amine was introduced to taurocholic acid sodium salt (Sigma, St. Louis, MO, USA) by activating the hydroxyl group using triethylamine (Sigma) and 4-nitrophenylchloroformate (Sigma). LMWH (Fraxiparin^®^, average molecular weight 4.5 kDa) was mixed with N-hydroxysuccinimide (NHS; Sigma), and then 1-ethyl-3-(3-dimethylaminopropyl) carbodiimidehydrochloride (EDAC; Sigma) was added for activation. Primary amines with taurocholic acid were conjugated on activated carboxylic group of LMWH via an EDAC/NHS coupling method.

### 2.2. Cell Lines

Human ductal pancreatic cancer cell lines PANC1 (KCLB 21469; Korea Cell Line Bank, Seoul, Republic of Korea) and MIA PaCa-2 (KCLB 21420; Korea Cell Line Bank, Seoul, Republic of Korea) are types of PDAC cell lines, and rat insulinoma cell line RINm (CRL-2057; ATCC, Manassas, VA, USA) is from PNETs (Appendix A). All cancer cell lines were cultured in Dulbecco’s modified Eagle’s medium (GenDEPOT, Katy, TX, USA) containing 10% fetal bovine serum (FBS; GenDEPOT) and 1% penicillin/streptomycin (PS). Human umbilical vein endothelial cells (HUVECs) were obtained from Lonza (C2517A; Allendale, NJ, USA) and cultured in endothelial growth medium (EGM-2 Bullet Kit, CC-3162; Lonza). HUVECs were used at passages 5 to 7. All cells were cultured at 37 °C in a humidified atmosphere of 95% air and 5% CO_2_.

### 2.3. LHT Effect on the Viability and Proliferation of Various Kinds of Pancreatic Cancer Cells

To evaluate the effect of LHT on the viability of pancreatic cancer cells, PANC1, MIA PaCa-2 and RINm cells (2 × 10^4^ cells per well) were seeded in 96-well flat-bottomed plates and cultured at 37 °C and 5% CO_2_ incubator for 24 h. The next day, cells were washed once with PBS and then further cultured with fresh culture medium containing different concentration of LHT (0, 50, 100 µg/mL) for 48 h. After that, the viability of the three types of pancreatic cancer cells were assayed using a Cell Counting Kit-8 (CCK-8; Dojindo, Rockville, Maryland, USA). The treated cells were rinsed with PBS and 100 µL of fresh culture medium was added with 10 µL CCK-8 per well. Plates were placed in a CO_2_ incubator for 4 h, and absorbance was measured with a microplate reader at 450 nm wavelength. Separately, the effect of LHT on the proliferation of pancreatic cancer cells was evaluated via anti-Ki67 immunostaining after cultivation without or with LHT (100 µg/mL) for 48 h. Simultaneously, DAPI (4′,6-diamidino-2-phenylindole), a blue, fluorescent DNA stain, was used as counter stain.

To further confirm whether or not LHT could affect the viability of pancreatic cancer cells (PANC1, MIA PaCa-2 and RINm cells), their levels of apoptosis were analyzed by flow cytometry using PE Annexin V Apoptosis Detection Kits (BD Pharmingen, Franklin Lakes, NJ, USA) after 48 h treatment of LHT. All flow cytometry analyses were carried out in triplicate using a FACS Calibur cell analyzer (Becton Dickinson, San Jose, CA, USA). Data acquisition and analysis were performed using Cell Quest Pro software (Becton Dickinson).

### 2.4. LHT Effect on VEGF Secretion of Various Pancreatic Cancer Cells

To evaluate whether LHT could affect the secretion of VEGF in pancreatic cancer cells, we measured the amount of VEGF secreted from them without or with LHT (100 µg/mL) treatment 24 h after cell seeding. After that, the cell lines were incubated in serum-free media (DMEM + 1% PS) for 6 h to undergo serum starvation. Then, VEGF in culture medium was measured using complete ELISA kits for human VEGF (KOMABIOTECH, Seoul, Korea) according to the manufacturer’s instructions. To evaluate whether LHT could affect the intracellular VEGF, cell lysates were extracted from pancreatic cancer cells using RIPA buffer (Thermo Fisher Scientific, Rockford, IL, USA) and proteinase inhibitor cocktail without or with LHT (100 µg/mL) treatment. Then, VEGF was measured using complete ELISA kits.

### 2.5. LHT Effect on the Viability and Proliferation of Human Umbilical Vein Endothelial Cells (HUVECs)

To evaluate the effect of LHT on the viability of HUVECs, they were seeded in 96-well flat-bottomed plates (2 × 10^4^ cells per well) and cultured at 37 °C and 5% CO_2_ incubator for 24 h. The next day, cells were washed once with PBS and then further cultured with fresh culture medium containing 10 ng/mL VEGF (recombinant murine VEGF_165_, VEGF-A; Peprotech, Rocky Hill, NJ, USA) and different concentration of LHT (0, 50, 100 µg/mL) for 48 h. After that, the viability of the HUVECs was assayed using a CCK-8 kit. Separately, the effect of LHT on the proliferation of HUVECs was evaluated via anti-Ki67 immunostaining after cultivation without or with LHT (100 µg/mL) for 48 h. Simultaneously, DAPI was used as counter stain. Furthermore, their apoptosis was further analyzed by flow cytometry using PE Annexin V Apoptosis Detection Kits after 48 h treatment of VEGF.

### 2.6. LHT Effect on Migration and Invasion of HUVECs In Vitro

To evaluate whether LHT could inhibit the migration and invasion activity of HUVECs, they were seeded in 6-well plates and cultured at 37 °C in a 5% CO_2_ atmosphere to full confluence [38,39]. After that, HUVECs were serum-starved for 6 h to inactivate them, and then plates were scratched with a 200-µL pipette tip. Cells were washed once with PBS, and then EBM-2 containing 1% FBS with VEGF (10 ng/mL) or VEGF with LHT (0, 50, 100 µg/mL) was added to each well. After 24 h treatment, the images of migrated cells were acquired by a Nikon digital camera (DXm1200c, Nikon Instruments Inc., Tokyo, Japan). Migrated cells were quantified using Image-Pro Plus 7.0 software (Media Cybernetics Inc., Rockville, MD, USA). Additionally, transwell invasion assays were performed using 6.5 mm diameter polycarbonate transwell plates (Corning Inc., Corning, NY, USA) with 8 µm-pore filters [40]. After serum-starving HUVECs for 6 h, the cells were suspended in 300 µL of EBM-2 containing 0.5% FBS and placed into upper chambers (1.3 × 10^5^ per well) coated with 100 μL Matrigel. To induce HUVEC invasion, lower chambers were filled with 500 µL of EBM-2 containing 0.5% FBS with VEGF (10 ng/mL) or VEGF with LHT (0, 50, 100 µg/mL). After 24 h treatment, nonmigrated cells were wiped away with cotton swabs and migrated cells were fixed with 4% paraformaldehyde and then stained with 0.05% crystal violet. Images were taken by inverted microscopy (Eclipse TE2000-S, Nikon, Tokyo, Japan), and the crystal violet-stained cells were quantified with Image-Pro Plus 7.0 software (Media Cybernetics).

### 2.7. LHT Effect on Tubular Formation of HUVECs In Vitro

To evaluate whether LHT could affect the angiogenic activity of HUVECs, in vitro endothelial tubular formation assay was conducted as described previously with some modifications [30,41]. Specifically, 200 µL of growth factor-reduced (GFR) Matrigel (BD Biosciences, San Jose, CA, USA) was pipetted into prechilled 24-well plates and polymerized for 1 h at 37 °C. Inactivated HUVECs (4 × 10^4^ cells/well) were seeded on the Matrigel and cultured in a final volume of 1 mL of EBM-2 containing 1% FBS with or without VEGF (10 ng/mL) and the different concentrations of LHT (0, 2, 5, and 10 µg/mL). After 18 h in 5% CO_2_ at 37 °C, HUVECs were photographed using a reverse phase-contrast photomicroscope (Nikon) and quantified using Image-Pro Plus 7.0 software (Media Cybernetics). On the other hand, rat aortic ring assays were performed with 48-well plates precoated with 100 µL of Matrigel and polymerized in a CO_2_ incubator for 1 h, as previously described [40,42]. Aortas isolated from 6-week-old male Sprague Dawley rats (Nara-Bio Company) were cleaned with PBS and cut into pieces with circumferences of 1 to 1.5 mm. Aortic rings were randomly placed in wells and overlaid with 100 µL of Matrigel. Then, the aortic rings were incubated in a final volume of 500 µL of EBM-2 medium with or without VEGF (10 ng/mL) and the different concentrations of LHT (0, 2, 5, and 10 µg/mL). The culture medium was changed every other day. After 6 days, sprouting microvessel growth was photographed using an inverted microscope (Nikon). The number of sprouting microvessels was determined with Image-Pro Plus 7.0 software (Media Cybernetics).

### 2.8. LHT Effect on Phosphorylation of Intracellular Signaling in HUVECs

To identify how LHT could affect the proliferation activity of HUVECs, the activations of intracellular signaling molecules were evaluated. After treatment of LHT (100 µg/mL) with or without VEGF (10 ng/mL) for 24 h, cell lysates were extracted from HUVECs using RIPA buffer (Thermo Fisher Scientific, Rockford, IL, USA) and proteinase inhibitor cocktail. Protein concentrations were quantified with BCA Protein Assay kits (Pierce). Then, Western blotting was carried out for checking the phosphorylation of VEGFR, ERK1/2 and FAK molecules with primary antibodies against ERK1/2 (Santa Cruz Biotechnology, Inc., Dallas, TX, USA), p-ERK1/2 (Santa Cruz Biotechnology), FAK (Signalway Antibody LLC., College Park, MD, USA), p-FAK (Signalway Antibody LLC), VEGFR (Abcam), p-VEGFR (Abcam) followed by incubation with horseradish peroxidase-conjugated secondary antibodies. All proteins were visualized using Supersignal West Pico chemiluminescent substrate (Pierce Biotechnology, Rockford, IL, USA). Signal intensities were quantitated by densitometry using Image-Pro Plus software (Media Cybernetics). Normalized values were expressed as the ratio of phosphoprotein/total protein levels.

### 2.9. Antitumor Effect of LHT on Orthotopic Pancreatic Tumors In Vivo

The animal study was carried out in compliance with the ARRIVE (Animal Research: Reporting of in vivo Experiments) guidelines. All procedures performed in this study were in accordance with the Hanyang University Animal Care and Use Committee Guidelines and were approved by the Institutional Animal Care and Use Committee (IACUC, 11-084A) of Hanyang University (Seoul, Korea). Orthotopic injections and implantations were carried out as previously described with minor modifications [43,44] (Appendix A). Male BALB/C nu/nu nude mice, 5 to 7 weeks old, were obtained from Nara-Bio (Seoul, Korea). Mice were anesthetized with Zoletil 50 (Virbac, France) and Rompun 10% (Bayer Korea, Korea) by intravenous injection (1 µL/g body weight). The three different human pancreatic cancer cell lines were detached from the plates, and cell numbers were counted using a hemocytometer. Pancreatic cancer cells (5 × 10^6^ cells/50 µL PBS) were injected into the head of pancreas organ with a 26 g needle. The needle was slowly withdrawn, and a cotton swab held in place for several seconds to prevent cell loss. After 2 weeks, to ensure the development of the cancer model, the same number of cancer cell lines was again injected under the same conditions. The mice were sacrificed 8 weeks after the second orthotopic injection. Tumors were dissected from the mice, and the outer parts of the tumor tissues were minced with a blade. Tumor lumps were weighed using an analytical balance (ML204, Mettler Toledo, Switzerland). The length of tumor lumps was calculated by measuring the length (L), width (W), and height (H) of each tumor lump with digital calipers (A&D Company, Limited, Tokyo, Japan). The equation of tumor volumes was calculated using following formula for a hemiellipsoid: tumor volume = 0.5236 × L × W × H. Then, tumor lumps with approximately 36 mg weight and 20 mm^3^ volume were trimmed and implanted into the heads of the pancreas organ in a new mouse. After surgery, mice were randomly divided into groups (n = 5). LHT was intravenously administered via the tail vein for 30 days (5 mg/kg/once every 2 days). PBS vehicle was administered to the control group. After 30 days, the mice were sacrificed and then tumor tissue in the pancreas organ was dissected, weighed, and measured for tumor volume.

For immunohistochemical staining, the dissected tumor tissue was infused with 30% sucrose in PBS overnight and embedded with optimum cutting temperature (OCT) compound (Tissuetech Inc., Miami, FL, USA) at −20 °C. Ten-micrometer sections were made using a cryosection microtome (Leica Microsystems Ltd., Nussloch, Germany) and kept in a sealed box at −20 °C until immunohistochemical staining. The sections were thawed at room temperature for 30 min, fixed in cold acetone for 15 min and dried at room temperature for 30 min. Serum blocking was carried out with nonspecific binding immunoglobulin in 20% goat serum in PBS-Tween 20 at room temperature for 30 min. To quantify cell proliferation and microvessel density, primary antibody (1:100 dilution) of anti-CD34 (rat antimouse IgG; Abcam) or anti-Ki67 (rabbit antimouse IgG; Abcam) were performed at room temperature for 1 h. After washing twice with PBS, secondary antibody (1:200 dilution) (FITC-conjugated goat antirat IgG; Abcam) was added and incubated at room temperature for 1 h. After washing twice with PBS, the sections were counterstained with DAPI-mounting medium (Vectashield H-1200, Vector Laboratories Inc., Burlingame, CA, USA). All images were obtained under fluorescence microscopy (Nikon).

### 2.10. Data Analysis

All data were expressed as means ± s.e.m. All graphs were plotted with SigmaPlot™ software (Systat Software Inc, San Jose, CA, USA) and statistical analysis was carried out using SigmaStat™ (Systat Software Inc). *p* < 0.05 was considered significant. All images were analyzed using Image-Pro Plus software (Media Cybernetics Inc., Rockville, MD, USA).

## 3. Results and Discussion

### 3.1. LHT Effect on Various Kinds of Pancreatic Cancer Cells

To investigate whether LHT could affect the viability and proliferation of pancreatic cancer cells, different concentrations of LHT were treated to pancreatic cancer cells such as PANC1, MIA PaCa-2 and RINm cells for 24 h (Figure 1A). PANC1 and MIA PaCa-2 are pancreatic ductal cancer cells from human PDAC, but RINm are endocrine beta cells from rat PNETs (Appendix A) [45]. In the cases of PANC1 and MIA PaCa-2 cells, their viabilities at treatment of 50 and 100 µg/mL of LHT were decreased by less than 10% compared to that of the no-treatment group. In the case of RINm cells, their viabilities at treatment of 50 and 100 µg/mL of LHT were 12–19% lower than the no-treatment group. On the other hand, even with LHT treatment, the viability of all groups was still twice higher than that of initial seeded cells (white bar). These results suggested that LHT could slightly attenuate cell proliferation but not significantly affect the viability of pancreatic cancer cells.

To clearly confirm whether LHT could inhibit cell proliferation rather than cell cytotoxicity, pancreatic cancer cells treated with 100 µg/mL of LHT were analyzed using flow cytometry for Annexin-V, which is commonly used as a marker of apoptosis (Figure 1B). The results showed that there were no differences between the no-treatment and the LHT-treated groups for all kinds of pancreatic cancer cells. Next, to further determine whether LHT could reduce proliferation of pancreatic cancer cells in vitro, each cell using anti-Ki67 antibody (a cell proliferation marker) was stained after treatment of LHT (100 µg/mL) for 48 h. The results showed that the number of Ki67-positive cells were decreased by treatment of LHT for all types of pancreatic cancer cells (Figure 1C). In addition, the cyclin D involved in regulating cell cycle progression was also weakly reduced in LHT-treated pancreatic cancer cells (Appendix A).

These results demonstrated that LHT could attenuate the proliferation of pancreatic cancer without cytotoxicity. Moreover, the inhibiting effect of LHT on the proliferation of RINm cells was stronger than those in the PANC1 or MIA PaCa-2 cells, which might be dependent on the characteristics of pancreatic cancer cells; PNETs have much abundant blood vessels in tumor tissue [46,47,48]. Therefore, we conducted the following studies to analyze the effect of LHT on the function of pancreatic cancer.

### 3.2. LHT Effect on VEGF Secretion of Various Pancreatic Cancer Cells

The amount of the representative growth factor, i.e., vascular endothelial growth factor (VEGF), secreted from pancreatic cancer cells was measured in terms of native activity of them. Interestingly, RINm cells secreted the highest amount of VEGF; its secretion level was almost 1.6 times higher than those of the PANC1 and MIA PaCa-2 cells (Figure 2A). In addition, the VEGF levels in the cell lysate of PANC1 and MIA PaCa-2 cells were slightly reduced, while the VEGF protein in the cell lysate of RINm cell was much more highly attenuated (Figure 2B). The reduced percentage of VEGF protein in PANC1, MIA PaCA-2, and RINm cells was about 4.3, 3.0, and 23.5%, respectively, compared to each control group (black bar). Based on these results, we found that LHT could effectively reduce VEGF content up to 6.5-fold more in the RINm cells than those in the other cells. In general, the extracellular signal-regulated kinase (ERK) is involved in functions including the regulation of mitosis in cells. ERK phosphorylation is an indicator of the mitogen-activated protein kinase (MAPK) signaling pathway that leads to the expression of the VEGF gene and promotes angiogenesis and vasculogenesis [49,50,51]. So, LHT might attenuate the proliferation of pancreatic cancer cells via inactivation of the MAPK/ERK signaling pathway, through blocking VEGF/VEGFR interaction [32,52,53]. For this reason, the fact that the secreted amount of VEGF differed depending on the pancreatic cancer cell type indicated that the reactivity of LHT would also be different. Therefore, we expected that LHT could have anticancer effects through antiangiogenesis in all types of cell lines, with more potent anticancer effects in RINm cells, specifically [52,53].

### 3.3. LHT Effect on the Viability and Proliferation of Human Umbilical Vein Endothelial Cells (HUVECs)

Usually, most cancer cells can secrete several growth factors such as VEGF, EGF and FGF molecules near the blood vessels [54,55,56]. Additionally, then, the endothelial cells in the blood vessels can be activated and proliferated to form new vessels. So, the effect of LHT on the endothelial cells were evaluated in the mimic microenvironment using the human umbilical vein endothelial cells (HUVECs) with VEGF-A (VEGF_165_) (10 ng/mL) (Figure 3A). Specifically, it is known that the VEGF family includes VEGF-A, -B, -C, and -D, and that type A is mainly secreted from solid cancer. VEGF-A binds to VEGF receptor 1 (VEGFR-1) and VEGF receptor 2 (VEGFR-2) to induce angiogenesis around tumor sites through autocrine processes. VEGFR-2 is a main signaling pathway for angiogenesis of endothelial cells. In pancreatic cancer, the binding of VEGF-A/VEGFR-2 causes cancer cell invasion and migration, resulting in accelerated tumor progression and poor prognosis [22]. For these reasons, recombinant VEGF-A (VEGF_165_) was used to clearly identify the effect of LHT on pancreatic cancer cells and endothelial cells.

The viability of HUVECs increased over 24 h without VEGF treatment. In the case of VEGF treatment alone, the cell viability was significantly further increased, which could be generally caused by activation of VEGFR on HUVECs via VEGF binding. Interestingly, when LTH was added to the VEGF-treated HUVECs, the cell viability decreased up to about 45% compared to the groups treated with VEGF alone. On the other hand, even with LHT treatment, the viability of all groups was still significantly higher than that of initial seeded cells (white bar). Based on these results, we found that LHT could attenuate the proliferation of HUVECs such as pancreatic cancer cells. To evaluate if this change in cell viability resulted from apoptotic effect of LHT, the LHT-treated HUVECs were analyzed using flow cytometry with the apoptosis marker annexin-V, which showed that about 3.9% of the HUVECs became apoptotic with LHT treatment (100 µg/mL) (red line, Figure 3B). This result appears to have some apoptosis in viability, but it was not statistically significant. Furthermore, to confirm whether LHT could attenuate the proliferation of HUVECs, HUVECs in all groups were immunostained with anti-Ki67 (Figure 3C). VEGF treatment increased the number of anti-Ki67-positive cells. However, LHT effectively attenuated the number of anti-Ki67-positive cells despite the treatment of VEGF. Collectively, these results indicated that LHT could hinder the proliferation of endothelial cells as well as pancreatic cancer cells, but this was not the cause of apoptosis, which has been a major problem with many other anticancer drugs [57].

### 3.4. LHT Effect on Migration, Invasion and Tubular Formation of HUVECs In Vitro

Angiogenesis is a complex and multistep process including migration, invasion and tube formation processes of endothelial cells [58,59]. Therefore, we evaluated whether or not LHT could affect VEGF-induced migration, invasion and tubular formation of HUVECs in vitro. LHT inhibited migration of HUVECs in a wound healing assay (Figure 4A,B). When 100 μg/mL of LHT was treated to HUVECs, their migration into the scratch area between yellow lines was strongly inhibited. In the invasion assay of HUVECs, VEGF could strongly induce their invasion for 24 h (Figure 4C,D). However, when accompanied with LHT and VEGF, invasion of HUVECs was completely arrested, like in the control group (no treatment of VEGF and LHT). Next, to assess the inhibitory effects of LHT on angiogenic activity of HUVECs, we treated HUVECs with different concentrations of LHT and measured tube lengths formed by HUVECs (Figure 5A,B). Compared to the control group, VEGF itself significantly increased the tube length of HUVECs. However, the tube lengths of HUVECs were significantly attenuated when the concentration of LHT was more than 5 μg/mL.

To confirm whether LHT could attenuate the migration, invasion, and tubular formation of HUVECs, ex vivo new vessels sprouting from aortic ring model were observed after treatment of VEGF alone or VEGF with LHT (Figure 5C,D). VEGF itself stimulated microvessel sprouting, leading to robust tubular formation. The number of sprouting microvessels decreased significantly with increasing concentrations of LHT. Based on these findings, we concluded that LHT could effectively inhibit the sprouting of microvessels from aortic rings ex vivo. In our previous study, we reported that LHT could block the fibroblast growth factor-2 (FGF-2) and platelet-derived growth factor-B (PDGF-B) in the endothelial cells [60]. In addition, we showed that LHT could inhibit multiple stages of angiogenesis, from initial response to maturation of the endothelial cells by pericyte coverage in vitro [60]. Collectively, our findings demonstrated that LHT could effectively inhibit the angiogenesis in and around tumor tissue.

### 3.5. LHT Effect on Intracellular Signaling Pathway of HUVECs In Vitro

To support LHT effect in a mechanistic way, we identified the phosphorylation of the VEGFR, extracellular signal-regulated kinase (ERK), and focal adhesion kinase (FAK) protein. As shown in Figure 6A **[61,62]**, the phosphorylation of VEGFR can stimulate the phosphorylation of ERK for cell migration and phosphorylation of FAK for cell migration. When LHT is added to HUVECs, it may attenuate the interaction between VEGF and VEGFR on the cellular membrane, thereby modulating the phosphorylation of ERK and FAK in the cytosol. To examine this, the phosphorylation of cell signaling molecules were analyzed using Western blots of the cell lysate of VEGF- or/and LHT-treated HUVECs (Figure 6B). In the case of VEGF-treated group, the bands of phosphor-VEGFR, phosphor-ERK1/2, and phosphor-FAK were strongly detected. In the case of the VEGF- and LHT-treated groups, however, the bands of phosphor-VEGFR, phosphor-ERK1/2, and phosphor-FAK declined by about 40, 55, and 35%, respectively (Figure 6C). Collectively, we found that the treated LHT could attenuate dephosphorylation of VEGFR on the cellular membrane of HUVECs. In addition, the FAK and ERK downstream of VEGFR signaling were further dephosphorylated, leading to the attenuation of migration, invasion, and tubular formation of HUVECs. Therefore, we expected that LHT could have anticancer effects through antiangiogenesis in HUVECs.

### 3.6. Antitumor Effect of LHT on Orthotopic Pancreatic Cancer Cells In Vivo

To investigate the effect of LHT on different types of pancreatic cancer cells in vivo, orthotopic pancreatic cancer models were used from different origins of pancreatic tumor cell lines (see Appendix A). To this end, we used a surgical protocol to make the orthotopic model, which mimicked the environment of natural pancreatic tumors (Appendix A) [63,64,65,66]. Briefly, pancreatic tumor cells were directly injected into the pancreas organ of mice. Later, we resected pancreatic tumors from the mice, cut them into equal parts of tumor lumps (volume 20 mm^3^ with 36 mg weight) and directly implanted them into the pancreas organ of a new mouse. This surgical strategy was used to create a complete orthotopic model without any metastasis [67]. After surgical implantation of the tumor lump, LHT (5 mg/kg/once every 2 days) was intravenously injected via tail vein for 30 days. The size of whole tumor tissue was dramatically attenuated after LHT administration for 30 days (Figure 7A). In addition, the overall size of RINm tumor tissue was relatively larger than other tumor tissues, which might be caused by high secretion of VEGF molecules from RINm cells (see Figure 2A). Although RINm cells could secrete larger amount of VEGF molecules, the injected LHT could effectively attenuate the growth of its tumor tissue in vivo. To quantify the effect of LHT on the orthotopic pancreatic tumor, the weight and volumes of the detached pancreatic tumor tissues were measured (Figure 7B,C). The average tumor weight of PANC1, MIA PaCa-2, and RINm pancreatic tumor in the LHT-treated group declined by 47, 41, and 49%, respectively, compared to the control (PBS injection) groups. In addition, the average tumor volume of PANC1, MIA PaCa-2, and RINm pancreatic tumor in the LHT-treated group decreased by 57, 45, and 57%, respectively, compared to the control groups. Furthermore, the body weight of the orthotopic pancreatic tumor-bearing mice was continuously increased, even though LHT was continuously administered (Appendix A). These data indicated that LHT could significantly and similarly inhibit tumor growth regardless of pancreatic cancer cell types in vivo. In that, LHT could have anticancer effects on different types of pancreatic cancer such as ductal adenocarcinoma (PANC1 and MIA PaCa-2) and pancreatic islet tumor cells (RINm).

To evaluate whether LHT attenuated the proliferation and angiogenesis in the orthotopic pancreatic tumor tissue, the sections of detached pancreatic tumor tissues were immunostained with anti-Ki67 and anti-CD34 antibody. In the case of LHT-treated group, the number of anti-Ki67-positive cells in the PANC1, MIA PaCa-2 and RINm cells was about 20, 28, and 12%, respectively, compared to each control group (green color, Figure 8A,B). On the other hand, the number of anti-CD34-positive cells in the PANC1, MIA PaCa-2 and RINm cells was about 35, 40, and 18%, respectively, compared to each control group (green color, Figure 8C,D). Based on these findings, we found that LHT could significantly reduce proliferation and angiogenesis in the orthotopic pancreatic tumor tissue in vivo, which was corelated to the result in vitro (see Figure 1, Figure 3 and Figure 5). Interestingly, RINm cells were much more sensitive to the injected LHT, while the two types of pancreatic cancer cells, i.e., PANC1, MIA PaCa-2, had relatively different responses. These results might be contributed to the characteristic of RINm cells showing higher secretion of VEGF. In fact, RINm cells secreted more VEGF proteins than the other cell lines [48]. Therefore, we predicted that this LHT could be much more strongly effective to PNETs exhibiting hypervascularity.

Previously, we reported that LHT could mainly affect angiogenic factors, such as VEGF and bFGF [31,60,68,69,70,71]. Higher effect of LHT in PNET cancer cells might be attributed to the interaction of the administered LHT with angiogenic factors. Therefore, if LHT is administered with cancer cells with hypervascularity, the anticancer effect of LHT will be very effective. On the other hand, dysfunctional vasculature in human pancreatic cancer as well as genetically engineered pancreatic animal models results in markedly reduced vascular density, which limits drug delivery to the pancreas [72]. For these reasons, a potential mechanism of resistance to VEGF inhibitors has been elucidated in pancreatic cancer models [73,74,75]. Indeed, by normalizing tumor vasculature, we can enhance drug delivery of VEGF inhibitors, thereby increasing chemotherapeutic activity [76]. For similar reasons, the effectiveness of Gemcitabine, the most effective anticancer medication against pancreatic cancer, may be minimal [77]. Therefore, the effect of LHT developed in an animal model of pancreatic cancer may be different in human clinical studies due to the different vascular density. Collectively, as a further study, it will be necessary to study the anticancer effect of LHT using a genetically engineered pancreatic animal model with reduced vascular density.

## 4. Conclusions

In this study, LHT could be used to treat all kinds of pancreatic cancer cells (i.e., PDACs and PNETs) in vitro and in vivo without cytotoxicity. Interestingly, LHT could be highly effective against RINm pancreatic cancer cells. In addition, LHT could reduce the migration, invasion, and tubular formation of endothelial cells (i.e., HUVECs), and inhibit sprouting of new microvessels from the aortic ring, which was attributed to attenuating the phosphorylation of VEGR on endothelial cells. These results were correlated to the fact that LHT suppressed tumor growth in both aggressive and malignant PDACs and hypervascularity-displaying PNECs in orthotopic models. Collectively, LHT could be a novel antipancreatic cancer medication that could overcome the typical shortcomings of chemotherapy, regardless of pancreatic cancer type.

## Figures and Tables

**Figure 1 cancers-13-05775-f001:**
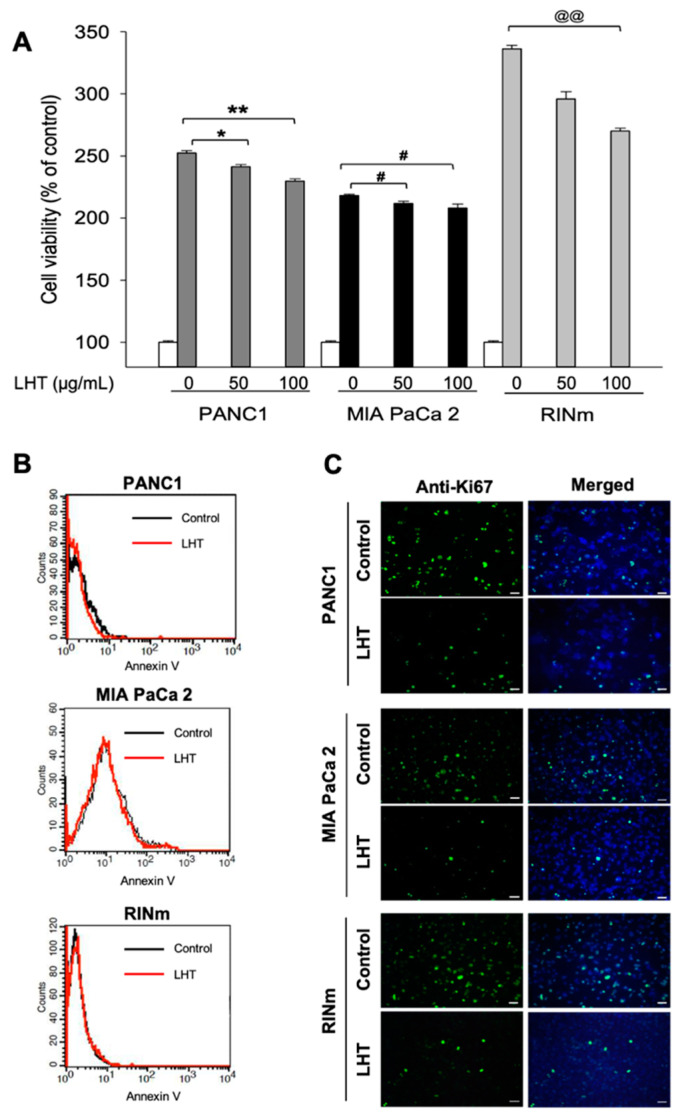
Proliferation inhibition of different types of pancreatic cancer cells after cultivation with LHT. (**A**) The viability of PANC1, MIA PaCa-2, and RINm cells after treatment of different concentrations (0, 50, 100 µg/mL) of LHT for 48 h. Data were expressed with mean ± s.e.m. (n = 8). White bar (control): the viability of each cells at seeding; Dark grey bar: PANC1 cell; Black bar: MIA PaCa-2 cell; Light grey bar: RINm cells. * *p* < 0.05, ** *p* < 0.01, ^#^ *p* < 0.05 and ^@@^ *p* < 0.01 versus the control for each group. (**B**) Flow cytometry analysis using PE Annexin-V Apoptosis Detection Kits (triplicate) after treatment of LHT (100 µg/mL) for 48 h. (**C**) Anti-Ki67 immunostaining of pancreatic cancer cells after treatment of LHT (100 µg/mL) for 48 h. Green: Anti-Ki67. Blue: DAPI. Scale bars indicate 100 µm.

**Figure 2 cancers-13-05775-f002:**
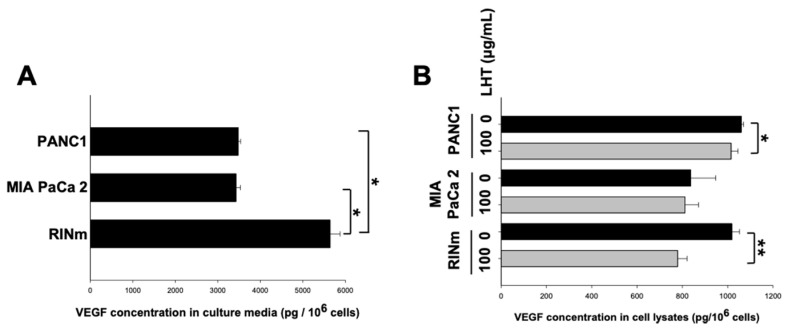
Reduction in VEGF secretion in pancreatic cancer cells after treatment of LHT. (**A**) Quantitative analysis of VEGF secretion from all types of pancreatic cancer cells without treatment of LHT. Data were expressed with mean ± s.e.m. (n = 5). * *p* < 0.05 versus the RINm cell line. (**B**) Relative quantification of VEGF in the cell lysate of pancreatic cancer cells after cultivation without (black bar) or with (grey bar) treatment of LHT (100 µg/mL) for 24 h. Data were expressed with mean ± s.e.m. (n = 5). * *p* < 0.05 and ** *p* < 0.01 versus each group without LHT.

**Figure 3 cancers-13-05775-f003:**
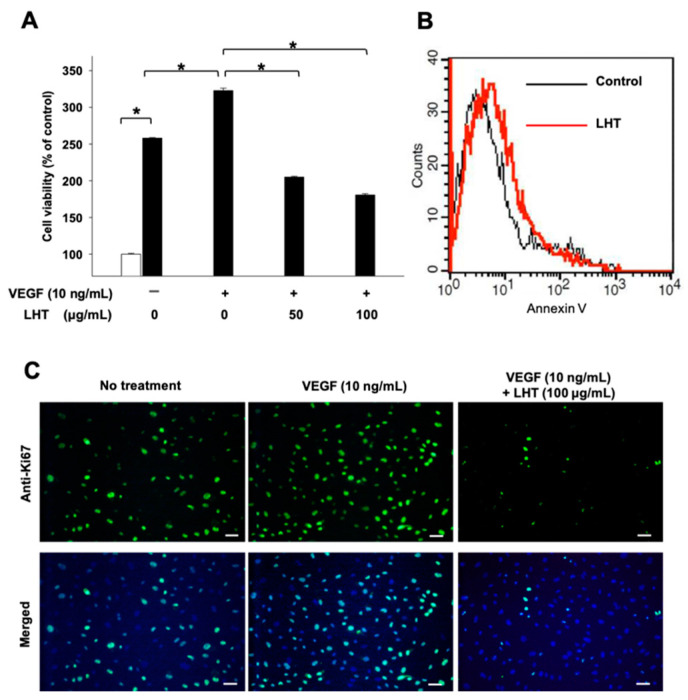
Inhibition of VEGF-induced proliferation of the human umbilical vein endothelial cells (HUVECs) after cultivation with LHT. (**A**) The viability of HUVECs after cultivation with different concentrations (0, 50, 100 µg/mL) of LHT plus VEGF (10 ng/mL) for 48 h. For control groups, the viability of HUVECs was measured after cell seeding or after 48 h of cultivation without anything. Data were expressed with mean ± s.e.m. (n = 5). * *p* < 0.05 at each comparison. (**B**) Flow cytometry analysis of HUVECs using PE Annexin V Apoptosis Detection Kits (triplicate) after treatment of LHT (100 µg/mL) for 48 h. Black line: no treatment of LHT. Red line: LHT treatment for 48 h. (**C**) Anti-Ki67 immunostaining of HUVECs without or with VEGF (10 ng/mL) and LHT (100 µg/mL). Blue: DAPI; Green: Anti-Ki67. Scale bars indicate 100 µm.

**Figure 4 cancers-13-05775-f004:**
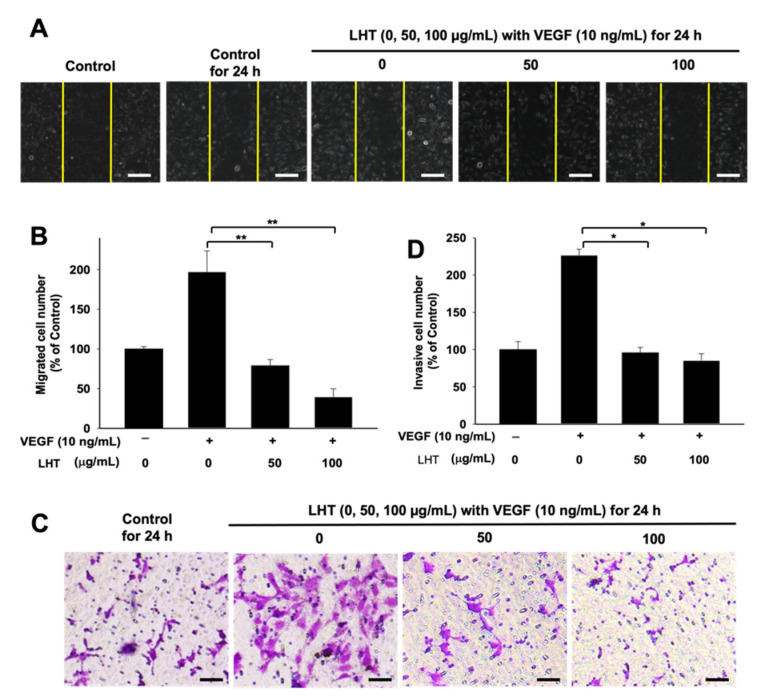
LHT effect on migration and invasion of HUVECs in vitro. (**A**) Optical image of migrating HUVECs into the scratched area after treatment of VEGF (10 ng/mL) or VEGF with LHT (0, 50, 100 µg/mL) for 24 h. Yellow line: scratched area at day 0. Scale bars indicate 200 µm. (**B**) Relative amount of the number of cells migrated into the scratched area, which was compared to that of control group (no treatment of both LHT and VEGF). Data were expressed with mean ± s.e.m. (n = 5). ** *p* < 0.01 versus the VEGF-treated group. (**C**) Optical image of crystal violet-stained HUVECs invaded through the transwell plate after cultivation with VEGF (10 ng/mL) or VEGF plus LHT (0, 50, 100 µg/mL) for 24 h. Scale bars indicate 200 µm. (**D**) Relative amount of the number of crystal violet-stained HUVECs invaded through the transwell plate. Data were expressed with mean ± s.e.m. (n = 5). * *p* < 0.05 versus the VEGF-treated group.

**Figure 5 cancers-13-05775-f005:**
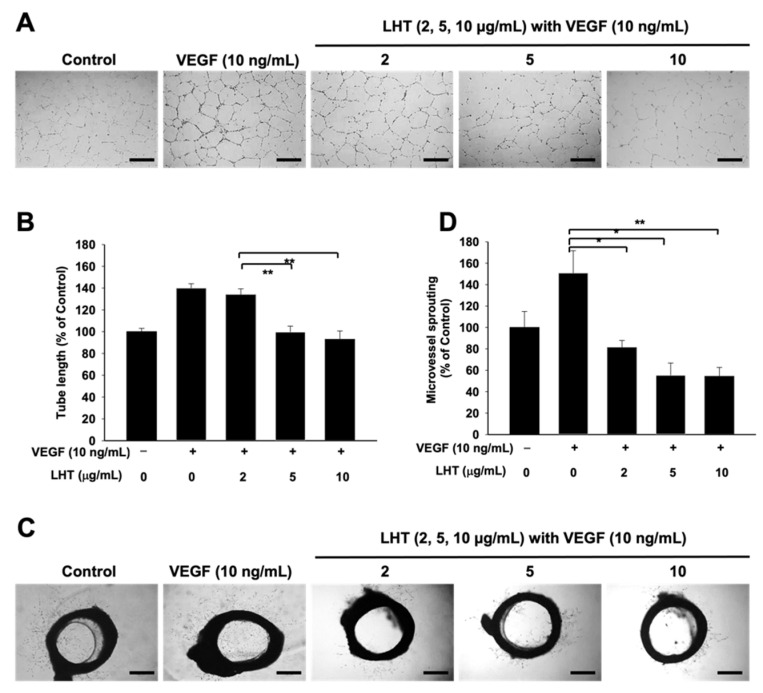
LHT effect on tubular formation of HUVECs in vitro. (**A**) Optical image of tubular formation of HUVECs after treatment of VEGF (10 ng/mL) or VEGF with LHT (0, 2, 5, 10 µg/mL) for 18 h. Scale bars indicate 500 µm. (**B**) Relative amount of tube lengths after treatment of VEGF (10 ng/mL) or VEGF with LHT (0, 2, 5, 10 µg/mL) for 18 h. Data were expressed with mean ± s.e.m. (n = 5). ** *p* < 0.01 versus the group treated with VEGF and LHT (2 µg/mL). (**C**) Optical images of new vessels sprouted from rat aortic ring after cultivation with VEGF (10 ng/mL) or VEGF with LHT (0, 2, 5, 10 µg/mL) for 6 days. Black ring: rat aorta ring. Scale bars indicate 500 µm. (**D**) Relative amount of the number of sprouting new vessels. Data were expressed with mean ± s.e.m. (n = 5). * *p* < 0.05 or ** *p* < 0.01 versus the VEGF-treated group.

**Figure 6 cancers-13-05775-f006:**
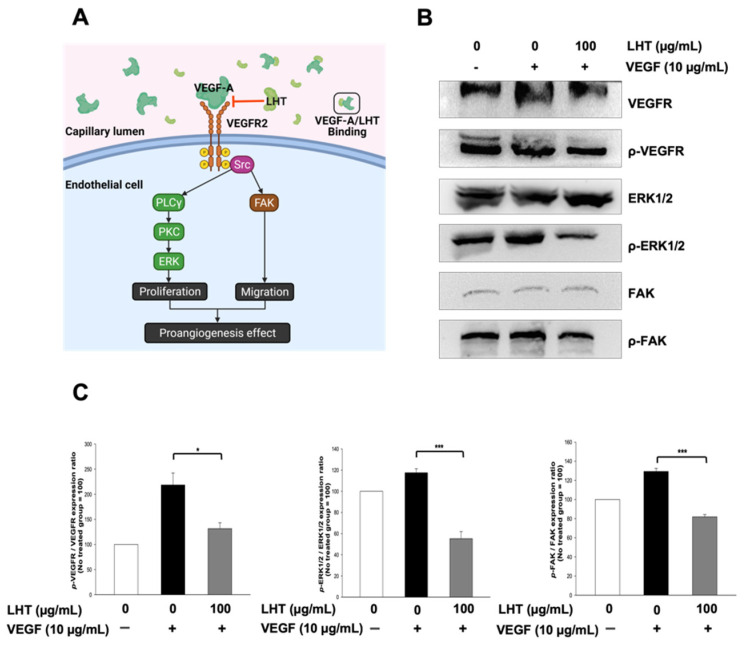
Inhibition effect of LHT on the cellular proliferation and migration through VEGF receptor, ERK and FAK pathways. (**A**) Multiple signaling pathways from the VEGF-A/VEGFR2 interaction to induce cellular migration and proliferation. LHT inhibits VEGF-A/VEGFR2 binding through VEGF-A/LHT interactions. (**B**) Western blot (cropped blots) of cell lysate of HUVECs after cultivation with VEGF (10 ng/mL) or VEGF with LHT (100 µg/mL) for 24 h. Full-length blots were presented in Appendix A. (**C**) Quantitative analysis of Western blotting bands cell lysate of HUVECs after cultivation with VEGF (10 ng/mL) or VEGF with LHT (100 µg/mL) for 24 h. Data were expressed with mean ± s.e.m. (n = 3). * *p* < 0.05, *** *p* < 0.001 versus the VEGF-treated group.

**Figure 7 cancers-13-05775-f007:**
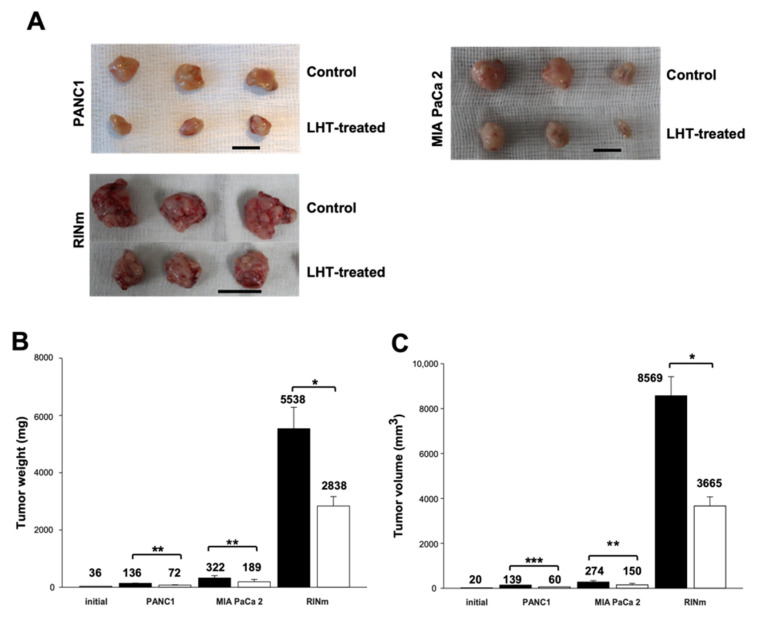
Inhibition of tumor growth and volume in an orthotopic pancreatic tumor during intravenous administration of LHT. (**A**) Optical images of solid tumors that were sacrificed and resected from the pancreas organ in the orthotopic pancreatic mouse after intravenous administration of LHT (5 mg/kg/once every 2 days) or PBS (control) for 30 days. Scale bars indicate 20 mm. (**B**) Tissue weight (mg) of tumor resected from control (black bar) or LHT-treated mice (white bar). Data were expressed with mean ± s.e.m. (n = 5). * *p* < 0.05, ** *p* < 0.01 versus each control groups. (**C**) Tissue volume (mm^3^) of tumor resected from control (black bar) or LHT-treated mice (white bar). Data were expressed with mean ± s.e.m. (n = 5). * *p* < 0.05, ** *p* < 0.01, *** *p* < 0.001 versus each control group.

**Figure 8 cancers-13-05775-f008:**
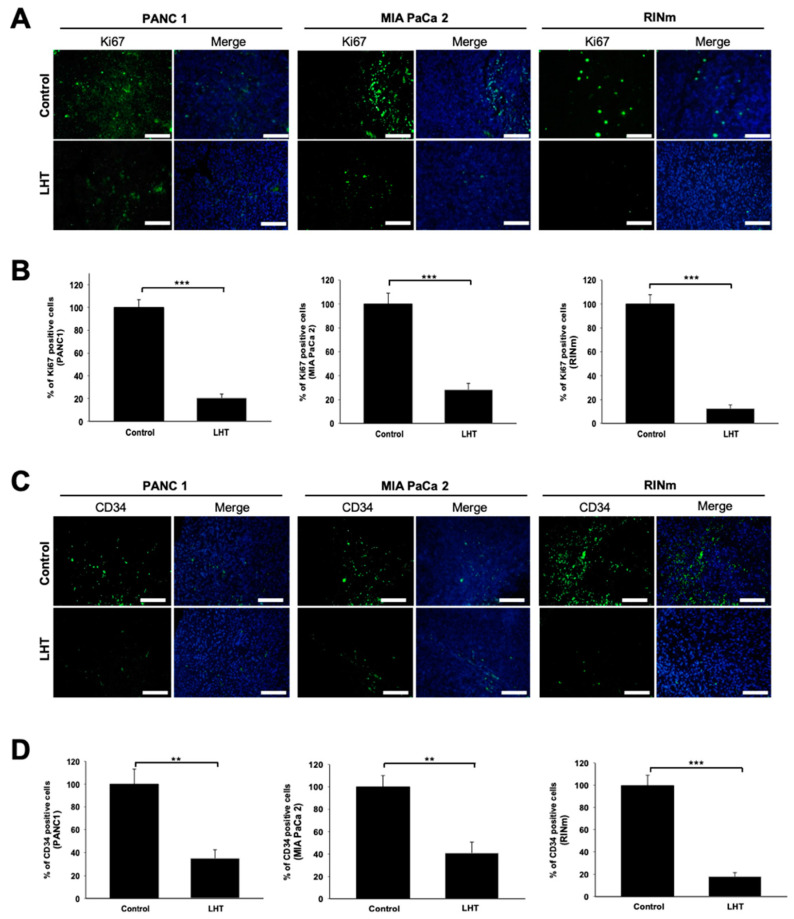
Inhibition of cell proliferation and angiogenesis in an orthotopic pancreatic tumor during intravenous administration of LHT. (**A**) Anti-Ki67 immunostaining of tumor tissues after intravenous administration of LHT (5 mg/kg/once every 2 days) or PBS (control) for 30 days. Blue: DAPI; Green: Anti-Ki67. Scale bars indicate 200 µm. (**B**) Relative quantification of anti-Ki67-positive cells in each group. Data were expressed with mean ± s.e.m. (n = 5). *** *p* < 0.001, compared to each control. (**C**) Anti-CD34 immunostaining of tumor tissues after intravenous administration of LHT (5 mg/kg/once every 2 days) or PBS (control) for 30 days. Blue: DAPI; Green: anti-CD34 microvessel. Scale bars indicate 200 µm. (**D**) Relative quantification of anti-CD34-positive cells. Data were expressed with mean ± s.e.m. (n = 5). ** *p* < 0.01, *** *p* < 0.001, compared to each control.

## Data Availability

Data is contained within the article or Appendix A.

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
