# Peer review of "Anticancer Effect of Heparin–Taurocholate Conjugate on Orthotopically Induced Exocrine and Endocrine Pancreatic Cancer"

_cancers, 2021, doi:10.3390/cancers13225775_

Round 1

Reviewer 1 Report

The authors present an interesting study that describes an investigation of the anti-cancer efficacy of LHT on pancreatic ductal adenocarcinoma (PDAC) and pancreatic neuroendocrine tumor (PNET), at orthotopic animal model.

This paper could be accepted, but some points have to be clarified or fixed before an affirmative action can be taken. Some minor revisions are required.

General comments:

- The background is not sufficient, and has critical flaws. To give a broad view, the authors in the introduction should discuss the efforts made to identify new molecules for the treatment of these aggressive and metastatic pancreatic cancer cell lines. Please provide some examples of anticancer agents in clinical use or others using Targeted approaches to improve outcomes of PDAC, but which nevertheless lead to the discovery of new ones.

Here are some examples:

  1. Drug Resistance Updates, 2021, 58, 100779; doi: https://doi.org/10.1016/j.drup.2021.100779
  1. Cancers, 2021, 13(17), 4389; doi: 10.3390/cancers13174389
  1. Pharmacol., 30 April 2020; doi: https://doi.org/10.3389/fphar.2020.00534

- Some research suggest that metastatic lesions are attenuated in modified heparin analogues containing mostly P-selectin inhibitory activity and in P-selectin-deficient mice. It could be interested to evaluate the LHT effect on P-selectin activity.

- In vivo, in vitro, and via should be italicized.

- All references should report the DOI number (for example in ref. 8, ref. 13, ref. 14 the DOI number is missing).

- A graphical abstract must be provided

Author Response

Reviewer #1:

The authors present an interesting study that describes an investigation of the anti-cancer efficacy of LHT on pancreatic ductal adenocarcinoma (PDAC) and pancreatic neuroendocrine tumor (PNET), at orthotopic animal model.

This paper could be accepted, but some points have to be clarified or fixed before an affirmative action can be taken. Some minor revisions are required.

General comments:

(Q1) The background is not sufficient, and has critical flaws. To give a broad view, the authors in the introduction should discuss the efforts made to identify new molecules for the treatment of these aggressive and metastatic pancreatic cancer cell lines. Please provide some examples of anticancer agents in clinical use or others using targeted approaches to improve outcomes of PDAC, but which nevertheless lead to the discovery of new ones.

Here are some examples:

  1. Drug Resistance Updates, 2021, 58, 100779; doi: https://doi.org/10.1016/j.drup.2021.100779
  1. Cancers, 2021, 13(17), 4389; doi: http://doi.org/10.3390/cancers13174389
  1. Pharmacol., 30 April 2020; doi: https://doi.org/10.3389/fphar.2020.00534

(Answer1) As you mentioned, we slightly modified the Introduction section including new drug development of pancreatic cancer.

Before correction in the Introduction section

Combinatorial chemotherapy with surgery or radiotherapy is a potential remedy to treat pancreatic cancer. Chemotherapy is used to treat progressive tumor cells located in pancreas because they remain without being completely removed by surgery. However, these strategies still have side effects such as hair loss, skin soreness and fatigue due to the damage of normal cells as well as cancer cells.

After correction in the Introduction section

Combinatorial chemotherapy with surgery or radiotherapy is a potential remedy to treat pancreatic cancer. Recently, glycogen synthase kinase 3 β (GSK3β) has emerged as a new potential target in PDAC due to its involvement in tumor-promoting property and chemoresistance [6-8]. To target GSK3β, its inhibitors such as ATP competitive and non-ATP competitive are developed [9-13] In addition, cyclin-dependent kinase 1 (CDK1), a stimulator of cell cycle, can be targeted to treat patients with PDAC [14]. To target PDAC diagnosis and therapy, a computational method of predicting the related gene expression and antidrug is also developed [15]. Chemotherapy is used to treat progressive tumor cells located in pancreas because they remain without being completely removed by surgery. However, these strategies still have side effects such as hair loss, skin soreness and fatigue due to the damage of normal cells as well as cancer cells.

(Q2) Some research suggest that metastatic lesions are attenuated in modified heparin analogues containing mostly P-selectin inhibitory activity and in P-selectin-deficient mice. It could be interested to evaluate the LHT effect on P-selectin activity.

- In vivo, in vitro, and via should be italicized.

- All references should report the DOI number (for example in ref. 8, ref. 13, ref. 14 the DOI number is missing).

- A graphical abstract must be provided

(Answer1) As you mentioned, we slightly modified the in vivo & in vitro as italic word. In addition, we checked and added the DOI number in all references. Also, a graphical abstract was added.

Before correction in the Introduction section

Heparin is commonly used as an anticoagulant, but it can be also used as an an-ti-cancer drug based on its anti-angiogenic properties [24]. The anti-angiogenic activity of heparin is attributed to its binding to VEGF and the subsequent inhibition of VEGF receptor (VEGFR) phosphorylation [25]. However, its dose should be critically monitored because its anticoagulant properties cause side effects such as hemorrhage and heparin-induced thrombocytopenia (HIT).

After correction in the Introduction section

Heparin is commonly used as an anticoagulant, but it can be also used as an an-ti-cancer drug based on its anti-angiogenic properties [24]. The anti-angiogenic activity of heparin is attributed to its binding to VEGF and the subsequent inhibition of VEGF receptor (VEGFR) phosphorylation [25]. In addition, heparin can significantly inhibit tumor cell adhesion through P-selectin mediation by the sulfate groups on its glucosamine residue [26,27]. However, its dose should be critically monitored because its anticoagulant properties cause side effects such as hemorrhage and heparin-induced thrombocytopenia (HIT).

Reviewer 2 Report

Thank you for the opportunity to review this paper. This is a in vitro and in vivo research which describe anti-cancer effect of heparin-taurocholate conjugate (LHT), which is one of the low molecular weight heparins (LMWH), against PDAC and PNET.

I think this manuscript has several serious problems to accept to this journal.

Comments

  1. Are there clinical data of LHT? It did not increase hemorrhagic complication? Pancreatic cancer and pancreatic surgery are highly associated with hemorrhagic complication. Moreover, History of antithrombotic agents use is a significant risk factor for postpancreatectomy hemorrhage.[1]
  2. From a clinical point of view, it is hard to believe that PDAC and PNET can be treated with the same drug, because the degree of malignancy is completely different.
  3. Author showed that LHT attenuated the production of VEGF through ERK dephosphorylation using HUVEC cell. However, a phase III trial already showed that the addition of bevacizumab (VEGF inhibitor) to chemotherapy does not improve survival in advanced pancreatic cancer patients.[2] From this point of view, the clinical effects of LHT are unfortunately not promising.
  4. Adverse effect of LHT in animal model should be indicated. Especially hemorrhagic complication.
  5. Similarly, survival of animal model should be investigated.

References

[1]        K. Nakamura et al., “Impact of Antithrombotic Agents on Postpancreatectomy Hemorrhage: Results from a Retrospective Multicenter Study,” J. Am. Coll. Surg., vol. 231, no. 4, pp. 460-469.e1, Oct. 2020, doi: 10.1016/j.jamcollsurg.2020.06.017.

[2]        H. L. Kindler et al., “Gemcitabine plus bevacizumab compared with gemcitabine plus placebo in patients with advanced pancreatic cancer: phase III trial of the Cancer and Leukemia Group B (CALGB 80303).,” J. Clin. Oncol., vol. 28, no. 22, pp. 3617–22, Aug. 2010, doi: 10.1200/JCO.2010.28.1386.

Author Response

Reviewer #2:

Thank you for the opportunity to review this paper. This is a in vitro and in vivo research which describe anti-cancer effect of heparin-taurocholate conjugate (LHT), which is one of the low molecular weight heparins (LMWH), against PDAC and PNET.

I think this manuscript has several serious problems to accept to this journal.

(Q1) Are there clinical data of LHT? It did not increase hemorrhagic complication? Pancreatic cancer and pancreatic surgery are highly associated with hemorrhagic complication. Moreover, History of antithrombotic agents use is a significant risk factor for postpancreatectomy hemorrhage.[1: K. Nakamura et al., “Impact of Antithrombotic Agents on Postpancreatectomy Hemorrhage: Results from a Retrospective Multicenter Study,” J. Am. Coll. Surg., vol. 231, no. 4, pp. 460-469.e1, Oct. 2020, doi: 10.1016/j.jamcollsurg.2020.06.017.]

(Answer) Recently, anticoagulant activity of heparin taurocholate was 12% when the equivalent amount of heparin itself was compared (Int. J. Cancer 124, 2755-2765, 2009, Biomaterials 33(17), 4424-4430, 2012). In addition, we carried out the safety study on intravenous infusion of LHT in preclinical models (SD-rat & beagle dogs) (Drug development and industrial pharmacy 42, 2016). There was no adverse effect of LHT infusion to beagle dog model. Furthermore, sulfated non-anticoagulant heparins (S-NACHs) might be preferred for potential clinical use in cancer patients without affecting hemostasis as compared to low molecular weight heparins (LMWHs) (Cancer Letter 250, 25-33, 2014). However, the hemorrhage complication of LHT should be carefully carried out at clinical level because there is no any clinical trials using LHT. So, we modified the Introduction section.

Before correction in the Introduction section

The conjugated taurocholate, one of bile acids, could not only reduce the anticoagulant activity of heparin, but also improve the stability of heparin itself via formation of pol-yproline-type helical structure [28, 32]. In addition, this LHT could be orally absorbable through the binding of taurocholate in the LHT on the bile acid receptors in the intestinal cells of preclinical animal models with higher safety [35, 36].

After correction in the Introduction section

The conjugated taurocholate, one of bile acids, could not only reduce the anticoagulant activity of heparin, but also improve the stability of heparin itself via formation of pol-yproline-type helical structure [28, 32]. In fact, the anticoagulant activity of LHT was 12% of unmodified heparin itself [32, 37]. Therefore, when LHT was intravenously infused in beagle dogs, there was no any safety issue [35, 36]. In addition, this LHT could be orally absorbable through the binding of taurocholate in the LHT on the bile acid receptors in the intestinal cells of preclinical animal models with higher safety [35, 36]. However, since clinical trials using this LHT have not yet been conduced, further investigation related to hemorrhage at the clinical level is needed [38]. 

(Q2) From a clinical point of view, it is hard to believe that PDAC and PNET can be treated with the same drug, because the degree of malignancy is completely different.

(Answer) As you mentioned, PDACs and PNETs have different malignancies in physiologically different patterns. As demonstrated in the results of our previous studies, each tumor expresses different amounts of VEGF, resulting in angiogenesis in different proportions. In this series of processes, the LHT used in this study does not directly act on PDAC and PNET cancer cells, but rather binds to the VEGF molecules secreted from them to exhibit anticancer effects. Indeed, based on the results of our previous studies, we reported that administered LHT affected angiogenic factors secreted by cancer cells rather than cancer cells themselves. Therefore, to explain these clearly, we modified the Results and Discussion section.

Before correction in Results and Discussion section

Interestingly, RINm cells were much sensitive to the injected LHT, while the two types of pancreatic cancer cells, i.e., PANC1, MIA PaCa-2, had relatively different responses to that. These results might be contributed to the characteristic of RINm cells showing higher secretion of VEGF. In fact, RINm cells secreted more VEGF proteins than the other cell lines [49]. Therefore, we predicted that this LHT could be much strongly effective to PNETs exhibiting hypervascularity.

After correction in Results and Discussion section

Interestingly, RINm cells were much sensitive to the injected LHT, while the two types of pancreatic cancer cells, i.e., PANC1, MIA PaCa-2, had relatively different responses to that. These results might be contributed to the characteristic of RINm cells showing higher secretion of VEGF. In fact, RINm cells secreted more VEGF proteins than the other cell lines [49]. Therefore, we predicted that this LHT could be much strongly effective to PNETs exhibiting hypervascularity.

Previously, we reported that LHT could mainly affect the angiogenic factors, such as VEGF and bFGF [32,54,70-73]. The reason that the administered LHT in this study was much effective in PNET cancer cells was thought to be the results obtained through the action of the administered LHT with angiogenic factors. Therefore, if LHT is administered with cancer cells having hypervascularity, the anticancer effect of LHT will be very ef-fective. On the other hand, dysfunctional vasculature in human pancreatic cancer as well as genetically engineered pancreatic animal model results in markedly reduced vascular density, which limits drug delivery to pancreas [74]. For these reasons, a potential mechanism of resistance to VEGF inhibitors has been elucidated in pancreatic cancer model [75-77]. Indeed, by normalizing tumor vasculature, it can enhance drug delivery of VEGF inhibitors, thereby increasing chemotherapeutic activity [78]. For similar reason, the effectiveness of Gemcitabine, the most effective anticancer medication to pancreatic cancer, may be minimal [79]. Therefore, the effect of LHT developed in an animal model of pancreatic cancer may be different in human clinical studies due to the different vascular density. Collectively, as a further study, it will be necessary to study the anti-cancer effect of LHT using a genetically engineered pancreatic animal model with reduced vascular density.

(Q3) Author showed that LHT attenuated the production of VEGF through ERK dephosphorylation using HUVEC cell. However, a phase III trial already showed that the addition of bevacizumab (VEGF inhibitor) to chemotherapy does not improve survival in advanced pancreatic cancer patients.[2: H. L. Kindler et al., “Gemcitabine plus bevacizumab compared with gemcitabine plus placebo in patients with advanced pancreatic cancer: phase III trial of the Cancer and Leukemia Group B (CALGB 80303).,” J. Clin. Oncol., vol. 28, no. 22, pp. 3617–22, Aug. 2010, doi: 10.1200/JCO.2010.28.1386.] From this point of view, the clinical effects of LHT are unfortunately not promising.

(Answer) Similar to human tumors, dysfunctional vasculature in genetically engineered pancreatic models results in markedly reduced vascular density, which limits drug delivery to pancreas. For these reasons, a potential mechanism of resistance to VEGF inhibitors has been elucidated in pancreatic cancer model. Indeed, by normalizing tumor vasculature, it can enhance drug delivery of VEGF inhibitors, thereby increasing chemotherapeutic activity. For similar reason, the effectiveness of Gemcitabine, the most effective anticancer medication to pancreatic cancer, may be minimal. Therefore, to explain these clearly, we modified the Results and Discussion section.

Before correction in Results and Discussion section

Interestingly, RINm cells were much sensitive to the injected LHT, while the two types of pancreatic cancer cells, i.e., PANC1, MIA PaCa-2, had relatively different responses to that. These results might be contributed to the characteristic of RINm cells showing higher secretion of VEGF. In fact, RINm cells secreted more VEGF proteins than the other cell lines [49]. Therefore, we predicted that this LHT could be much strongly effective to PNETs exhibiting hypervascularity.

After correction in Results and Discussion section

Interestingly, RINm cells were much sensitive to the injected LHT, while the two types of pancreatic cancer cells, i.e., PANC1, MIA PaCa-2, had relatively different responses to that. These results might be contributed to the characteristic of RINm cells showing higher secretion of VEGF. In fact, RINm cells secreted more VEGF proteins than the other cell lines [49]. Therefore, we predicted that this LHT could be much strongly effective to PNETs exhibiting hypervascularity.

Previously, we reported that LHT could mainly affect the angiogenic factors, such as VEGF and bFGF [32,54,70-73]. The reason that the administered LHT in this study was much effective in PNET cancer cells was thought to be the results obtained through the action of the administered LHT with angiogenic factors. Therefore, if LHT is administered with cancer cells having hypervascularity, the anticancer effect of LHT will be very ef-fective. On the other hand, dysfunctional vasculature in human pancreatic cancer as well as genetically engineered pancreatic animal model results in markedly reduced vascular density, which limits drug delivery to pancreas [74]. For these reasons, a potential mechanism of resistance to VEGF inhibitors has been elucidated in pancreatic cancer model [75-77]. Indeed, by normalizing tumor vasculature, it can enhance drug delivery of VEGF inhibitors, thereby increasing chemotherapeutic activity [78]. For similar reason, the effectiveness of Gemcitabine, the most effective anticancer medication to pancreatic cancer, may be minimal [79]. Therefore, the effect of LHT developed in an animal model of pancreatic cancer may be different in human clinical studies due to the different vascular density. Collectively, as a further study, it will be necessary to study the anti-cancer effect of LHT using a genetically engineered pancreatic animal model with reduced vascular density.

(Q4) Adverse effect of LHT in animal model should be indicated. Especially hemorrhagic complication. Similarly, survival of animal model should be investigated.

(Answer) As you mentioned, the evaluation of the survival rate of experimental animals following LHT drug administration is also a very important part in terms of drug efficacy evaluation. Unfortunately, this experiment could not be performed in this study due to the limitations of the experimental animal design. Therefore, in a future research plan, we will separately evaluate the survival rate of experimental animals following LHT administration. On the other hand, the evaluation of the side effects of LHT was not also performed in this study. However, we already showed the results of previous studies on the toxicity evaluation of LHT using beagle dogs, which was answered in (Q1).

Before correction in the Introduction section

The conjugated taurocholate, one of bile acids, could not only reduce the anticoagulant activity of heparin, but also improve the stability of heparin itself via formation of pol-yproline-type helical structure [28, 32]. In addition, this LHT could be orally absorbable through the binding of taurocholate in the LHT on the bile acid receptors in the intestinal cells of preclinical animal models with higher safety [35, 36].

After correction in the Introduction section

The conjugated taurocholate, one of bile acids, could not only reduce the anticoagulant activity of heparin, but also improve the stability of heparin itself via formation of pol-yproline-type helical structure [28, 32]. In fact, the anticoagulant activity of LHT was 12% of unmodified heparin itself [32, 37]. Therefore, when LHT was intravenously infused in beagle dogs, there was no any safety issue [35, 36]. In addition, this LHT could be orally absorbable through the binding of taurocholate in the LHT on the bile acid receptors in the intestinal cells of preclinical animal models with higher safety [35, 36]. However, since clinical trials using this LHT have not yet been conduced, further investigation related to hemorrhage at the clinical level is needed [38]. 

Round 2

Reviewer 2 Report

The revised manuscript was corrected well.